# Pix2Plan: A Set Prediction Approach for End-to-End Wireframe Parsing using Two-Level Polygon Queries

## Abstract

Extracting accurate wireframes of built environments from remotely sensed data is essential for several tasks, such as urban reconstruction, mapping, indoor floorplan extraction, and building roof extraction. Despite significant progress in the area, extracting accurate tight-layout wireframes from remotely sensed data remains an open problem. In this paper, we introduce Pix2Plan, a single-stage end-to-end set prediction transformer for wireframe parsing using two-level polygon queries and junction matching. Pix2Plan employs a DETR-style encoder-decoder transformer to predict a set of two-level polygon queries and a global set of junction vertices. The polygon vertex proposals are matched to the predicted junctions in the scene to obtain a wireframe as a planar graph. Thus, Pix2Plan can retrieve the building roof / indoor room polygons in the wireframe in a tight layout. Evaluation on several challenging planar graph datasets demonstrates that Pix2Plan achieves state-of-the-art performance across precision, recall, and shape quality metrics while exhibiting high efficiency.

## 1 Introduction

Extracting accurate wireframes of built environments from remotely sensed data is an essential task in scene understanding, building roof modelling, indoor floorplan extraction, etc. Developing automatic methods to extract building wireframes and indoor floorplans is crucial since these are difficult to obtain and laborious to annotate manually. Such methods have several applications in downstream tasks such as urban reconstruction, architecture, cultural heritage digitization, and building information modeling.

Recently, several works in computer vision address wireframe parsing, and these methods can be broadly grouped as (i) graph-based and (ii) polygon-based methods. Graph-based methods follow a bottom-up approach, e.g., extracting floorplans by detecting a set of corner vertices and line segments connecting the vertices (Liu et al., 2018; Jiacheng Chen, 2022; Zorzi & Fraundorfer, 2023). This results in a planar graph whose vertices, edges, and faces represent the corners, walls/ridges, and rooms/roof segments of the floorplan/roof wireframe, respectively. Such bottom-up graph learning approaches have the advantage of being able to directly predict tight-layout wireframes as planar graphs from which both line-level and polygon-level components can be inferred. However, a disadvantage of such methods is that they are often characterized by complex architectures and multi-stage training pipelines. Furthermore, these methods can be sensitive to the noisy nature of the input data formats, such as 3D point clouds, depth maps, and density images, leading to incomplete wireframe predictions.

Polygon-based methods on the other hand approach this as a top-down detection problem (Chen et al., 2019; Stekovic et al., 2021; Yue et al., 2023; Liu et al., 2025; Xu et al., 2024; Su et al., 2023), i.e., they model the task as detecting a set of separate room/roof-level polygons which are then aggregated to form the final floorplan/roof wireframe. These top-down methods have the advantage of being able to leverage powerful and robust transformer architectures that excel at detection tasks (Carion et al., 2020; Zhu et al., 2021). However, top-down approaches have the major limitation of not being able to directly make wall/ridge-level predictions. The walls/ridges in the floorplans/roof wireframe are simply obtained from the edges of the room/roof polygons, leading to duplicate wall/ridge segments.

Obtaining tight layout floorplans and roof wireframe from such polygons often requires non-trivial, heuristic post-processing that may be error-prone.

In this paper, we propose **Pix2Plan**, an end-to-end attention-based approach that extracts polygonal wireframes from 3D LiDAR scans as planar graphs where face polygons and junctions are predicted using multi-level queries. Our method adopts the two-level queries approach of recent polygon-based methods (Yue et al., 2023; Liu et al., 2025) to predict polygons while also following the planar graph learning approach of bottom-up methods (Jiacheng Chen, 2022). This allows Pix2Plan to have strong polygon-level detection performance while also being able to extract high-quality wireframes in a tight layout without complex post-processing.

We evaluate our method on three challenging wireframe parsing datasets: (i) the Building 3D dataset (Wang et al., 2023), (ii) the Structured 3D floorplan dataset (Zheng et al., 2020), and (iii) the SceneCAD floorplan dataset (Avetisyan et al., 2020). Pix2Plan outperforms state-of-the-art methods for building roof extraction task and achieves competitive performance with state-of-the-art methods for the floorplan extraction task, while ensuring tight layouts in the predicted wireframes.

In summary, we make the following novel contributions in this paper:

1. We propose Pix2Plan, a deep neural network that can predict high-quality building roofs and indoor floorplans as planar graphs while maintaining high-detection performance.

2. We propose a novel scene-level junction decoder capable of predicting junction coordinates of the roof/room polygons in the scene. The polygon vertices predicted by the network are aligned to these predicted junctions to ensure completeness in the predicted planar graph.

Comprehensive evaluations across three challenging datasets demonstrate that Pix2Plan is capable of predicting accurate, high-quality, tight layout wireframes in an end-to-end fashion.

## 2 RELATED WORKS

Floorplan reconstruction is the task of extracting accurate indoor floorplans in vector data formats from remotely sensed data such as indoor LiDAR scans, density maps, multi-view images, etc. Similarly, building roof extraction is the task of predicting accurate building roof wireframes from LiDAR scans, aerial images, etc. Several early methods use a host of classical image processing and graph-based techniques to extract such wireframes from remotely sensed data (Adan & Huber, 2011; Lladós et al., 1997; Monszpart et al., 2015; Ikehata et al., 2015). More recently, deep learning has been applied successfully in place of classical methods. Recent wireframe parsing methods can be broadly grouped into (i) graph-based and (ii) polygon-based methods.

**Graph-based methods** tend to work in a bottom-up fashion where building roofs and indoor floorplans are modeled as planar graphs and neural networks are employed to learn the vertices and edges of these planar graphs. FloorNet (Liu et al., 2018) uses a triple-branch pipeline to directly consume multimodal inputs, followed by an integer programming formulation to reconstruct floorplans from intermediate predictions. In Floorplan-Jigsaw (Lin et al., 2019), the authors estimate initial local floorplans from a set of partial scene inputs, followed by posed optimal placement and refinement to reconstruct the complete floorplan. Among recent wireframe detection methods, HAWP (Xue et al., 2020; 2023) employs vertex and edge detection modules followed by matching between the predicted vertices and edges to reconstruct the final indoor scene wireframe as a planar graph. Similarly, in HEAT (Jiacheng Chen, 2022), the authors propose an end-to-end pipeline that first detects all corners in the input image and uses transformer decoders to detect if an edge is present for all possible corner pairs.

Bottom-up methods can directly predict wireframes as tight-layout planar graphs, with rich part-level information. Such floor plans are more suitable for downstream CAD workflows, requiring accurate edge/wall level annotations. However, a common limitation of these approaches is that they often require extensive post-processing to retrieve the final planar graph from the intermediate part-level predictions. Also, the quality of the final planar graphs can be adversely affected by undetected corners or edges.

**Polygon-based methods**, on the other hand, reconstruct indoor floorplans by directly predicting roof/room-level polygons in a top-down fashion without any intermediate steps. In Floor-SP (Chen et al., 2019), the authors use a standard Mask-RCNN (He et al., 2017) to detect room segments from a top-down density image of indoor RGBD scans, followed by a room-wise coordinate descent to optimize the vectorized room polygon coordinates. Finally, Floor-SP adopts a heuristic-based post-processing simplification to recover the final vector floorplans. MonteFloor (Stekovic et al., 2021) also uses a Mask-RCNN detector to detect room segment proposals from a top-down density image, followed by a Monte Carlo Tree Search to select the true positive room segments. The room segments are then refined using a metric network that measures how well the predicted floor plan fits the input density map.

More recently, attention-based transformer architectures have shown great potential for several vision tasks such as object detection (Carion et al., 2020; Zhu et al., 2021), instance segmentation (Cheng et al., 2021; Xie et al., 2021), tracking (Meinhardt et al., 2022), etc. Also, DETR-style networks (Carion et al., 2020; Zhu et al., 2021) have also shown significant potential for several primitive detection tasks such as line segment detection (Xu et al., 2021), plane detection (Tan et al., 2021), etc. Due to the simple training objective of polygon-based top-down methods, recent approaches have successfully leveraged efficient DETR-style architectures for floorplan reconstruction, resulting in a much simpler model architecture and training pipeline compared to previous heuristic-based top-down methods. In RoomFormer (Yue et al., 2023), the authors use multi-level polygon queries to predict room polygons as a set of ordered sequences of vertices. The multi-level queries used in this approach allow RoomFormer to use a DETR-style decoder to predict arbitrary-length room polygons. In PolyRoom (Liu et al., 2025), the authors use an alternate uniform sampling representation where each room polygon query has a fixed number of vertices. These initial polygon queries obtained from a pre-trained segmentation network are then passed to a DETR-style decoder to obtain the refined room polygons. Finally, non-corner vertices are suppressed to obtain the final simplified floor plans.

Polygon-based floorplan extraction methods predict floorplans as a set of room polygons, allowing them to leverage simpler and robust DETR-style transformer architectures. However, in such polygon-based methods, the lack of edge-level reasoning results in errors at room corners and walls. Also, these methods still require heuristic-based post-processing to recover tight-layout floorplans, which is error-prone.

More recently, some **hybrid approaches** have also been proposed to address some of these afore-mentioned limitations. SLIBO-Net (Su et al., 2023) uses a slicing-box representation for floorplans, allowing for efficient representation of complex floorplans. The authors employ a room center regression network and a slicing box detection network, from which the final floorplans are recovered by clustering the slicing boxes based on the predicted room centers. Also, in FRI-Net (Xu et al., 2024), the authors represent each room polygon as a set of parametric lines and use an encoder-decoder network to predict the occupancy of a set of query points. The occupancy codes of parametric lines are merged to recover room polygons and, in turn, the complete floor plan.

**Training strategy.** In addition to the above classification, present state-of-the-art methods for building roof and floor plan extraction can be further categorised based on the training pipeline as (i) single-stage methods and (ii) two-stage methods. Methods such as HAWP (Xue et al., 2020; 2023), HEAT (Jiacheng Chen, 2022), and RoomFormer (Yue et al., 2023) can be categorised as **single-stage methods** since these methods can be trained in an end-to-end fashion without the need for any pretraining steps. On the other hand, **two-stage methods** involve an extensive pretraining stage to generate initial wireframe/floorplan proposals, which are then refined by the second stage of training. In FRI-Net (Xu et al., 2024), the authors use a pretrained RoomFormer (Yue et al., 2023) to generate initial room polygon proposals, which are further refined as parametric lines. Similarly, PolyRoom (Liu et al., 2025) uses a pretrained segmentation network to predict initial room polygons, which are also refined in the second stage of training. Single-stage methods have the advantage of being much more efficient to train and extend to new tasks. Two-stage methods, on the other hand, trade off the extensibility of the models for higher performance on a specific task.

Compared with these previous methods, Pix2Plan makes a step change and reconstructs building roofs and floorplans in a single-stage hybrid fashion, combining the advantages of both bottom-up and top-down methods while maintaining high polygon-level performance.

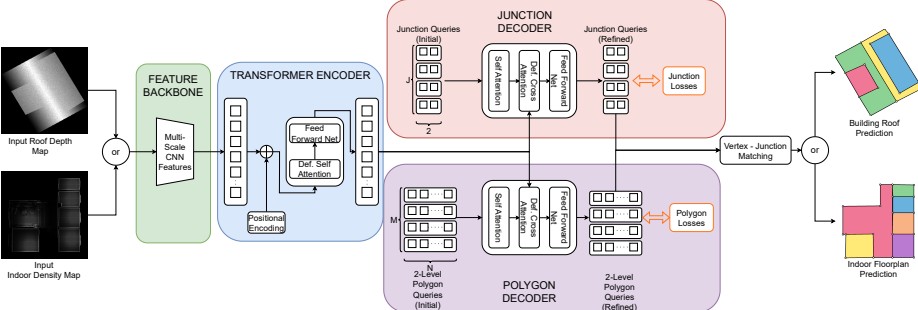

Figure 1: **Overall architecture of Pix2Plan.** The input to the networks are either top-down depth maps for building roof extraction or density maps from terrestrial LiDAR for indoor floorplan extraction. The input density or depth map is passed to a ResNet50 (He et al., 2016) feature backbone followed by a transformer encoder. The encoded sequence features are then passed to dedicated polygon and junction decoders with corresponding polygon and junction queries respectively. The polygon/junction decoders predict polygon/junction vertex coordinates. The predicted polygon and junction vertices are then passed to the Vertex-Junction matching module to predict high-quality tight-layout building roof and indoor floorplan wireframes.

## 3 METHODOLOGY

### 3.1 WIREFRAME REPRESENTATION

Extracting indoor floor plans and roof wireframes from remotely sensed data is, in essence, a detection problem. Depending upon the learning approach used, wireframes are commonly represented in two ways in recent literature.

Bottom-up and graph-based methods treat wireframes as a combination of geometric primitives, i.e., corners, edges, and polygons (Liu et al., 2018; Lin et al., 2019; Jiacheng Chen, 2022). Such methods can extract wireframes as a planar graph that can be readily used in downstream CAD applications. However, these methods are also characterized by multi-stage training pipelines and heuristic postprocessing. On the other hand, top-down polygon-based methods treat wireframes as simply a set of polygons and do not directly represent the edges and corners in the scene (Yue et al., 2023; Liu et al., 2025). Instead, they rely on the polygon vertices to implicitly encode edge and corner information. This allows these methods to leverage simple and robust architectures since the problem is essentially reduced to predicting a set of room/roof polygons in a scene. However, these methods have the limitation of not being able to predict walls/edges in the wireframe directly. To overcome this, these methods employ heuristic-based postprocessing to recover edges from the negative spacing between predicted room polygons, which is error-prone.

In our proposed method, we represent a wireframe in a bottom-up fashion, i.e., as a combination of polygons and corners. However, to overcome the limitations of bottom-up representations, we employ the two-level queries proposed in (Yue et al., 2023) to ensure dense supervision for the primitive detection tasks. Thus, we employ dedicated decoders for polygons & corner junctions and treat each task as a primitive set prediction task.

### 3.2 PIX2PLAN: ARCHITECTURE

We illustrate the overall architecture of Pix2Plan in Figure 1. First, for floor plans, the indoor scene point cloud is projected from the top-down view to a fixed image size, resulting in the input density map, which is the input to the network. Similarly, top-down depth maps are generated for the building roofs from the building point clouds. Next, a CNN backbone extracts multi-scale features from the density/depth map. These features are aggregated, patched, and passed to a transformer encoder along with positional information to obtain refined features. Then, two dedicated transformer decoders, namely the Junction Decoder and the Polygon Decoder, detect corner junctions and room/roof-segment polygons in the scene, using two-level queries. Finally, the polygon vertices are matched to

the predicted set of junctions to obtain the indoor floor plan and building roof wireframe in a planar graph format.

**Feature Backbone:** We use a standard ResNet50 (He et al., 2016) network to extract pixel-level features from the input density/depth map $I_d \in \mathbb{R}^{H \times W}$. Following (Zhu et al., 2021), we use the multi-scale feature maps from all $l$ layers of the backbone to ensure learning of robust local and global level features for accurate prediction of vertices. Then, we flatten all feature maps and add sinusoidal positional encodings. These flattened feature maps with positional information are then passed to the transformer encoder.

**Transformer Encoder:** The multi-scale feature maps with positional encodings from the CNN backbone are passed to a transformer encoder to compute the refined features. As in (Zhu et al., 2021), we use a multi-scale deformable self-attention module and a feed-forward network in each encoder layer.

**Polygon Decoder:** For predicting the roof faces/room polygons from the encoded features, we use a transformer decoder where each layer consists of a self-attention module, a multi-scale deformable cross-attention module, and a feed-forward network. Inspired by (Yue et al., 2023), the polygon queries are modeled as two-level polygon queries with one level representing the complete set of polygons in the scene and the other level directly representing each polygon's vertices. The two-level polygon queries are of the form $P_Q \in \mathbb{R}^{M \times N \times 2}$, where $M$ is the maximum number of polygons in a scene and $N$ is the number of vertices in each polygon. The final decoder layer predicts both the polygon vertex coordinates and a class label $c$ for each vertex, indicating if a vertex is valid.

**Junction Decoder:** In addition to the polygon decoder, we use a dedicated transformer decoder to detect the corner junctions of walls/roof segments in the scene from the encoded features. The Junction decoder uses the same type of decoder layers as in the Polygon decoder. For decoding junctions, we use a standard junction query representation $J_Q \in \mathbb{R}^{J \times 2}$. The final decoder layer predicts both the junction coordinates and class labels for each junction.

**Vertex-Junction Matching:** Once both sets of polygons and junctions in an image are predicted, we perform a distance-based matching between the junctions and polygon vertices. For each polygon vertex, we match it to the nearest corner in the set of junctions predicted by the junction decoder. This results in a tight layout roof/floorplan wireframe as a planar graph. It can be noted that Pix2Plan can recover tight-layout roof wireframes and floorplans directly from the input density/depth images without the need for any heuristic post-processing steps.

## 3.3 LOSSES

We use the standard set prediction loss formulation used in DETR-style networks (Carion et al., 2020; Zhu et al., 2021), where each prediction is optimally matched to the ground truth objects via the Hungarian matching algorithm to compute losses.

**Polygon Losses.** Once the two-level polygon predictions from the polygon decoder are optimally matched with the ground truth polygons, we compute the classification, coordinates, and rasterized polygon losses. For classifying each label, we use the binary cross-entropy loss:

$$\mathcal{L}_{P_{\text{cls}}}^m = -\frac{1}{N} \sum_{n=1}^{N} c_n \cdot \log(\hat{c_n}) - (1 - c_n) \cdot \log(1 - \hat{c_n}) \tag{1}$$

where $c_n$ and $\hat{c_n}$ are the ground truth and predicted polygon vertex logits. $N$ is the total number of polygon vertices. Non-matched vertices are assigned to the no-object class in the above equation.

Similarly, we use the L1 distance error as the loss for regressing the polygon vertex coordinates:

$$\mathcal{L}_{P_{\text{coord}}}^m = \frac{1}{N_m} \mathbb{1}_{\{m \le M^{\text{gt}}\}} d(P_m, \hat{P}_m) \tag{2}$$

where $N_m$ is the maximum number of vertices in the ground truth polygons and $M^{\text{gt}}$ is the number of ground-truth polygon instances. Also, $d$ measures the L1 distance between the ground truth polygon vertices $P_m$ and the predicted polygon vertices $\hat{P}_m$, which is sliced to the same length as $P_m$.

Additionally, we also compute the Dice loss between the rasterized predicted polygons and ground truth polygon masks as follows:

$$\mathcal{L}_{P_{\text{ras}}}^m = \mathbb{1}_{\{m \leq M^{\text{gt}}\}} \text{Dice}(R(P_m), R(\hat{P}_m)) \tag{3}$$

where $R(.)$ is the differentiable rasterizer (Lazarow et al., 2022) used to rasterize a given polygon.

**Junction Losses.** In addition to the above polygon losses, we also supervise the junction decoder with the junction classification and junction coordinates regression losses. We use a standard binary cross-entropy loss for classifying each junction logit:

$$\mathcal{L}_{J_{\text{cls}}}^m = -\frac{1}{N_j} \sum_{n=1}^{N_j} c_{n_j} \cdot \log(\hat{c_{n_j}}) - (1 - c_{n_j}) \cdot \log(1 - \hat{c_{n_j}}) \tag{4}$$

where $c_{n_j}$ and $\hat{c}_{n_j}$ are the ground truth and predicted junction vertex logits. $N_j$ is the maximum number of junction vertices in a scene.

Similar to the polygon coordinate regression, we also use the L1 distance loss for regressing junction coordinates:

$$\mathcal{L}_{J_{\text{coord}}}^m = \mathbb{1}_{\{m \leq M_j^{\text{gt}}\}} d(J_m, \hat{J}_m) \tag{5}$$

where $M_j^{\text{gt}}$ is the number of ground truth junction instances. $J_m$ and $\hat{J}_m$ are the ground truth and predicted junction coordinates, respectively. The total loss is then calculated as:

$$\mathcal{L} = \lambda_{P_{\text{cls}}} \mathcal{L}_{P_{\text{cls}}}^m + \lambda_{P_{\text{coord}}} \mathcal{L}_{P_{\text{coord}}}^m + \lambda_{P_{\text{ras}}} \mathcal{L}_{P_{\text{ras}}}^m + \lambda_{J_{\text{cls}}} \mathcal{L}_{J_{\text{cls}}}^m + \lambda_{J_{\text{coord}}} \mathcal{L}_{J_{\text{coord}}}^m \tag{6}$$

## 4 Experiments

In this section, we report the quantitative and qualitative results of evaluations conducted on the Building3D (Wang et al., 2023), Structured3D (Zheng et al., 2020), and the SceneCAD (Avetisyan et al., 2020) datasets.

**Implementation Details** The input density/depth images were resized to $256 \times 256$. We use a ResNet-50 backbone as in HEAT (Jiacheng Chen, 2022). For the transformer encoder and decoder in Pix2Plan, we used 6 layers each. The deformable attention module contains 8 heads and 4 feature levels. We employ the Adam optimizer with a weight decay of $1 \times 10^{-4}$. For the loss weights, we used $\lambda_{P_{\text{cls}}} = 2$, $\lambda_{P_{\text{coord}}} = 5$, $\lambda_{P_{\text{ras}}} = 1$, $\lambda_{J_{\text{cls}}} = 2$, and $\lambda_{J_{\text{coord}}} = 5$. Following (Yue et al., 2023), we used an initial learning rate of $2 \times 10^{-4}$ for the Building3D and Structured3D experiments. For the SceneCAD experiments, we used an initial learning rate of $5 \times 10^{-5}$. Pix2Plan was trained for 100 epochs on the Building3D dataset, 500 epochs on the Structured3D dataset, and 400 epochs on the SceneCAD dataset. Following (Yue et al., 2023), we decay the learning rate by a factor of 0.1 after 80% epochs. All experiments were conducted on a single RTX A5000 GPU with 24GB of VRAM.

### 4.1 Datasets

**Building3D** The Building3D dataset (Wang et al., 2023) is an urban-scale dataset consisting of aerial LiDAR point clouds of nearly 160,000 buildings spanning across 16 cities in Estonia. Each building point cloud is accompanied by high-quality 3D meshes and wireframe models. The dataset contains buildings across 60 different roof types, allowing for high intra-class variance. In our experiments, we used the Tallinn subset of the dataset, consisting of 32,618 training samples and 3,472 test samples. Since the official test samples do not have publicly available annotations, we further divided the official train split into 80/10/10% splits for 26,093 training, 3,261 validation, and 3,263 testing samples, respectively. For each building sample, we generated top-down building roof depth maps by projecting and rasterizing the 3D wireframe, which serve as the model inputs for the task of building roof wireframe extraction.

**Structured3D** Structured3D (Zheng et al., 2020) is a large-scale dataset consisting of indoor LiDAR scans of 3500 houses. We adopt the official splits of 3000 training samples, 250 validation samples, and 250 testing samples. We follow the procedure adopted by (Jiacheng Chen, 2022; Stekovic et al., 2021; Yue et al., 2023) to convert the registered multi-view RGB-D scans to top-down density images of size $256 \times 256$.

Table 1: Quantitative evaluation results on Building3D (Wang et al., 2023) dataset. **Bold** and underlined scores indicated best and second-best scores respectively.

| Type | Method | Roof | | | | Corner | | | Angle | | |
|---|---|---|---|---|---|---|---|---|---|---|---|
| | | IoU | Prec. | Rec. | F1 | Prec. | Rec. | F1 | Prec. | Rec. | F1 |
| Single stage | RoomFormer (Yue et al., 2023) | 59.9 | 42.1 | 56.5 | 48.2 | 41.2 | 55.4 | 47.2 | 36.1 | 47.0 | 40.8 |
| | Pix2Plan (ours) | **65.0** | 45.4 | **66.6** | **54.0** | **43.8** | **65.6** | **52.5** | **38.7** | **54.9** | **45.4** |
| Two stage | FRI-Net (Xu et al., 2024) | 34.5 | **58.3** | 44.4 | 50.4 | 19.7 | 30.6 | 23.9 | 4.9 | 7.3 | 5.9 |
| | PolyRoom (Liu et al., 2025) | 62.6 | 42.0 | 58.9 | 49.1 | 40.1 | 58.4 | 47.6 | 35.0 | 48.7 | 40.7 |

**SceneCAD** SceneCAD (Avetisyan et al., 2020) is a collection of real-world RGB-D scans consisting of 3D room layout annotations. The layout annotations are converted to floorplan polygons using the procedure adopted by (Yue et al., 2023). Again, we follow the same preprocessing steps as in Structured3D to convert the RGB-D scans to top-down density images.

## 4.2 EVALUATION METRICS

Following (Jiacheng Chen, 2022; Stekovic et al., 2021; Yue et al., 2023), we report precision, recall, & F1 scores at three different levels: roof/room polygons, corners, and angles. For each roof/room polygon in the ground truth, we iterate over the predictions, retrieve the predicted polygon with the best IoU, and compute the above metrics.

## 4.3 RESULTS

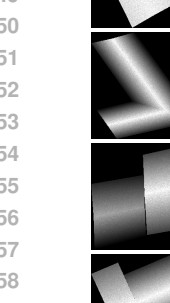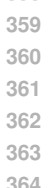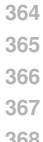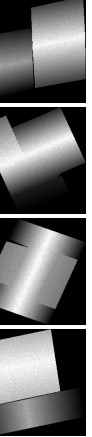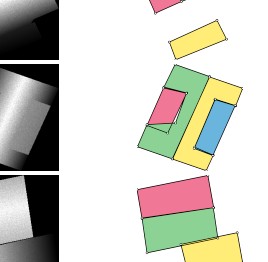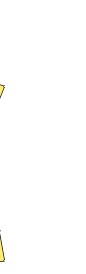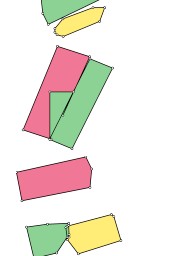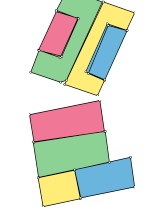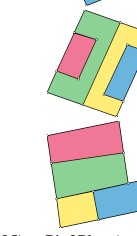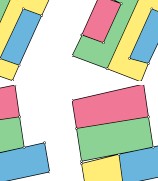

Depth Map    RoomFormer (Yue et al., 2023)    FRI-Net (Xu et al., 2024)    PolyRoom (Liu et al., 2025)    Pix2Plan (ours)    Ground Truth

Figure 2: **Qualitative Comparisons on the Building3D test set (Wang et al., 2023).** Roof segment colors are assigned randomly and do not indicate any class labels. Building roof wireframes predicted by existing methods tend to be characterised by errors such as missing roof segments, overlapping polygons and redundant vertices. FRI-Net in particular struggles due to diagonal edges dominating this dataset. However, Pix2Plan is free of such errors and is able to consistently predict accurate building roof wireframes in a tight-layout.

**Comparisons** In this section, we present the quantitative and qualitative comparisons between Pix2Plan and other state-of-the-art methods. In Table 1 we present the quantitative results of experiments on the Building3D dataset. It can be seen that Pix2Plan achieves state-of-the-art

Table 2: **Quantitative evaluation results on Structured3D (Zheng et al., 2020) dataset.** Results of previous works are taken from (Yue et al., 2023; Xu et al., 2024; Liu et al., 2025; Su et al., 2023).

| Type | Method | Room Prec. | Room Rec. | Room F1 | Corner Prec. | Corner Rec. | Corner F1 | Angle Prec. | Angle Rec. | Angle F1 |
|---|---|---|---|---|---|---|---|---|---|---|
| | HAWP (Xue et al., 2020) | 77.7 | 87.6 | 82.3 | 65.8 | 77.0 | 70.9 | 59.9 | 69.7 | 64.4 |
| | LETR (Xu et al., 2021) | 94.5 | 90.0 | 92.2 | 79.7 | 78.2 | 78.9 | 72.5 | 71.3 | 71.9 |
| Single stage | HEAT (Jiacheng Chen, 2022) | 96.9 | 94.0 | 95.4 | 81.7 | 83.2 | 82.5 | 77.6 | 79.0 | 78.3 |
| | RoomFormer (Yue et al., 2023) | 97.9 | 96.7 | 97.3 | 89.1 | 85.3 | 87.2 | 83.0 | 79.5 | 81.2 |
| | Pix2Plan (ours) | 97.0 | 95.8 | 96.4 | 87.8 | 84.1 | 85.9 | 76.1 | 73.0 | 74.5 |
| | Floor-SP (Chen et al., 2019) | 89. | 88. | 88. | 81. | 73. | 76. | 80. | 72. | 75. |
| | MonteFloor (Stekovic et al., 2021) | 95.6 | 94.4 | 95.0 | 88.5 | 77.2 | 82.5 | 86.3 | 75.4 | 80.5 |
| Two stage | FRI-Net (Xu et al., 2024) | 99.5 | 98.7 | 99.1 | 90.8 | 84.9 | 87.8 | 89.6 | 84.3 | 86.9 |
| | SLIBO-Net (Su et al., 2023) | 99.1 | 97.8 | 98.4 | 88.9 | 82.1 | 85.4 | 87.8 | 81.2 | 84.4 |
| | PolyRoom (Liu et al., 2025) | 98.9 | 97.7 | 98.3 | 94.6 | 86.1 | 90.2 | 89.3 | 81.4 | 85.2 |

Table 3: **Quantitative evaluation results on SceneCAD (Avetisyan et al., 2020) dataset.**

| Type | Method | Room IoU | Corner Prec. | Corner Rec. | Corner F1 | Angle Prec. | Angle Rec. | Angle F1 |
|---|---|---|---|---|---|---|---|---|
| | HEAT (Jiacheng Chen, 2022) | 84.9 | 87.8 | 79.1 | 83.2 | 73.2 | 67.8 | 70.4 |
| Single stage | RoomFormer (Yue et al., 2023) | 91.7 | 92.5 | 85.3 | 88.8 | 78.0 | 73.7 | 75.8 |
| | Pix2Plan (ours) | 90.9 | 89.8 | 84.2 | 86.9 | 72.4 | 68.8 | 70.5 |
| | Floor-SP (Chen et al., 2019) | 91.6 | 89.4 | 85.8 | 87.6 | 74.3 | 71.9 | 73.1 |
| Two stage | FRI-Net (Xu et al., 2024) | 92.3 | 92.8 | 85.9 | 89.2 | 78.3 | 73.6 | 75.9 |
| | PolyRoom (Liu et al., 2025) | 92.8 | 96.8 | 86.1 | 91.2 | 81.7 | 74.5 | 78.0 |

performance for the task of building roof extraction. This is also observed in the qualitative results presented in Figure 2, where it can be observed that building roof predictions of RoomFormer (Yue et al., 2023) is characterized by missing roof segment polygons and topological errors such as overlapping polygons. FRI-Net (Xu et al., 2024) on the other hand struggles greatly with predicting non-axis aligned polygon edges. This is due to the room-wise decoder favoring axis-aligned lines instead of diagonal lines in the prediction. PolyRoom (Liu et al., 2025), while exhibiting better performance, still suffers from errors such as redundant vertices, overlapping roof segments and gaps between polygons. Pix2Plan, on the other hand, is free from all these errors and is able to consistently predict accurate tight-layout building roof wireframes.

For the task of indoor floorplan extraction, Pix2Plan performs competitively among the single-stage methods. However, two-stage methods tend to outperform single-stage methods. This is observed in the results of experiments on the Structured3D and SceneCAD datasets presented in Tables 2 and 3, respectively. Still, it can be observed that Pix2Plan is capable of predicting high-quality indoor floorplans in a tight layout as seen in Figures 3 and 4. It can be observed again that RoomFormer (Yue et al., 2023) predictions are characterized by angular errors and redundant vertices. Again, it can be seen that FRI-Net (Xu et al., 2024) struggles especially with diagonal walls. Furthermore, all present state-of-the-art methods are characterised by gaps between adjacent walls, resulting in a non-tight layout of indoor floorplans. Once again, Pix2Plan is able to predict accurate tight-layout indoor floorplans without exhibiting any of the aforementioned errors.

In general, it is observed that Pix2Plan outperforms state-of-the-art methods for building roof extraction while closely matching the performance of SOTA single-stage methods for indoor floorplan extraction. We posit that the hybrid approach of matching polygon vertices to junction coordinates adopted by Pix2Plan is better suited for wireframe datasets such as Building3D. In floorplan datasets such as Structured3D and SceneCAD, there is a higher emphasis on room-level detection performance and shape quality of predicted polygons and a lesser emphasis on the planarity of the predicted floorplan graph. Despite this, Pix2Plan matches state-of-the-art single-stage methods, indicating Pix2Plan as a viable method capable of predicting tight-layout wireframes that can be further finetuned by future two-stage methods for better detection performance.

**Limitations** While Pix2Plan achieves state-of-the-art performance on several challenging datasets, it is also characterized by some limitations. Specifically, the primary cause of failure for Pix2Plan is the prediction of incomplete polygons or completely missing polygons. Examples of such failure cases are presented in the supplementary. This is a contributing factor to the lower performance compared to two-stage methods. One way this could be further improved is by adopting some of the

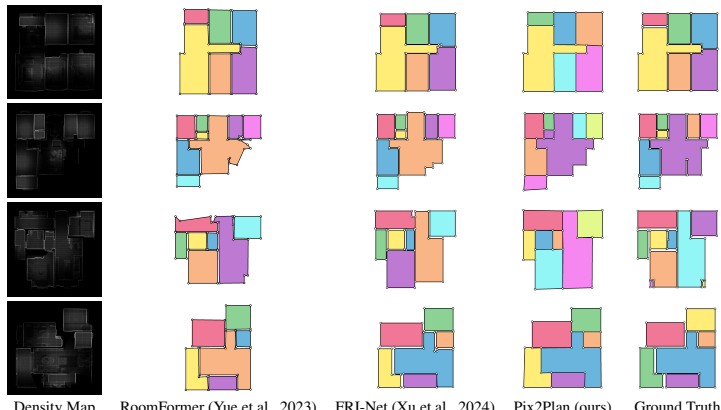

Figure 3: **Qualitative Comparisons on the Structured3D test set (Zheng et al., 2020).** Room colors are assigned randomly and do not indicate any class labels. Existing methods tend to predict indoor floorplans with gaps between walls along with errors such as incorrect walls and missing room polygons. FRI-Net struggles with diagonal walls due the assumption that axis-aligned walls dominating floorplan datasets. Pix2Plan is able to predict accurate indoor floorplans in a tight-layout without gaps between walls.

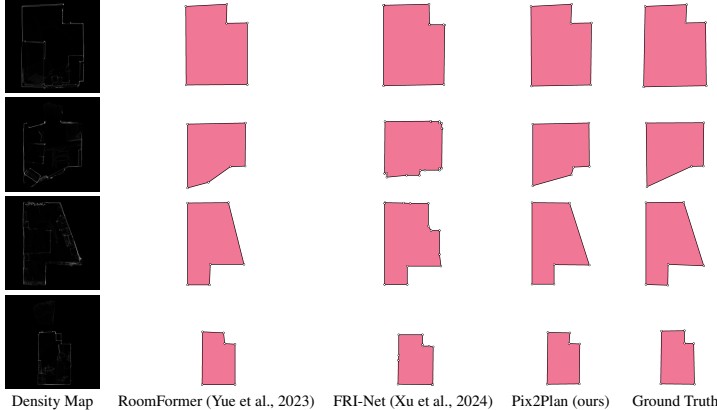

Figure 4: **Qualitative Comparisons on the SceneCAD validation set (Avetisyan et al., 2020).** Roof segment colors are assigned randomly and do not indicate any class labels.

pretraining strategies of two-stage methods, such as using the pretrained room-wise encoder as in FRI-Net (Xu et al., 2024) or the room query initialization module used in PolyRoom (Liu et al., 2025) to obtain high-quality initial polygon proposals. On the other hand, the relatively dated ResNet-50 (He et al., 2016) backbone could be replaced by more modern alternatives such as attention-based vision transformer (Dosovitskiy et al., 2021) or ConvNext (Liu et al., 2022) backbones for richer feature representations. Addressing these limitations is presently the focus of ongoing research. Nevertheless, Pix2Plan achieves state-of-the-art performance for building roof extraction and matches other state-of-the-art single-stage methods for indoor floorplan extraction.

## 5 CONCLUSION

In this paper, we present Pix2Plan, an end-to-end single-stage deep neural network capable of predicting high-quality building roof and indoor floorplan wireframes from 3D scans of buildings and indoor environments. By adopting a hybrid approach of predicting both polygons as well as the junctions in the scene, Pix2Plan can ensure the tight layout of predicted wireframes through a polygon vertex to junction matching step. This allows Pix2Plan to outperform state-of-the-art methods on the task of building roof extraction and match state-of-the-art single-stage methods for indoor floorplan extraction across several instance and polygon quality metrics.

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

## A  APPENDIX

In this appendix, we present the results of the ablation experiments conducted for the junction decoder and vertex-junction matching of Pix2Plan in Section B. We demonstrate some examples of the common failure cases of Pix2Plan in Section C. Finally, we present some additional qualitative examples for Pix2Plan in Section D.

## B  ABLATION EXPERIMENTS

In this section, we report the quantitative and qualitative results of the ablation experiments for the novel Junction Decoder and Vertex-Junction Matching used in the Pix2Plan architecture. To ablate the junction decoder of the Pix2Plan architecture, we entirely remove the junction decoder and the corresponding junction losses from the pipeline. We simply use the polygons predicted by the polygon decoder for performing evaluations. This configuration is referred to as 'Pix2Plan w/o JD' in Table 4 and Figure 5. On the other hand, to ablate the vertex-junction matching step of the architecture, we keep the junction decoder and simply skip the vertex-junction matching step during inference, using the polygons from the polygon decoder for evaluations. This configuration is referred to as 'Pix2Plan w/o JM' in Table 4 and Figure 5. The full Pix2Plan architecture, including both the Junction Decoder and the Vertex-Junction Matching step, is referred to as 'Pix2Plan (full)'.

From the results in Table 4, it can be observed that both 'Pix2Plan w/o JD' and 'Pix2Plan w/o JR' perform worse than the proposed 'Pix2Plan (full)' configuration. This can be attributed to the fact that both the junction decoder and vertex-junction matching step are essential to recover tight-layout building roof wireframes. It should also be noted that when Pix2Plan is trained with the junction decoder and the vertex-junction matching step is skipped during inference, the performance is slightly worse than when removing the junction decoder altogether. This is probably due to the multi-task learning objective of learning both junctions and polygons in a scene causing the polygon decoder to perform slightly worse. Nonetheless, the full Pix2Plan architecture with both the junction decoder and the vertex-junction matching step outperforms the ablation baselines across all metrics. Thus it is clear that both the junction decoder and the vertex-junction matching step are essential for Pix2Plan's state-of-the-art performance.

In Figure 5, we present some examples of building roof predictions from the various configurations of Pix2Plan used in the ablation experiments. It can be seen that the predictions made by 'Pix2Plan w/o JD' and 'Pix2Plan w/o JM' are both characterized by gaps between roof segments and excess vertices. 'Pix2Plan (full)' on the other hand is able to consistently predict tight-layout building roof wireframes.

## C  FAILURE CASES

In Figure 6, we demonstrate some examples of Pix2Plan's failure cases across all datasets. It can be seen that the most common mode of failure is when the network partially or completely misses roof/room polygons. The case of completely missing polygons can be attributed to the polygon decoder while the case of missing vertices or junctions could be due to the performance bottleneck in either the polygon decoder or the junction decoder. Although these errors indicate a detection bottleneck in the polygon and junction decoders, it should be noted that similar modes of failure

Table 4: **Quantitative results of the ablation experiments on Building3D (Wang et al., 2023) dataset.** 'Pix2Plan w/o JD' denotes the version of Pix2Plan without the Junction Decoder. 'Pix2Plan w/o JM' denotes the version of Pix2Plan without the Vertex-Junction Matching step. 'Pix2Plan (full)' is the full architecture with both the Junction Decoder and Vertex-Junction Matching step. **Bold** and underlined scores indicate best and second-best scores respectively.

| Method | Roof | | | | Corner | | | Angle | | |
|---|---|---|---|---|---|---|---|---|---|---|
| | IoU | Prec. | Rec. | F1 | Prec. | Rec. | F1 | Prec. | Rec. | F1 |
| Pix2Plan w/o JD | 64.7 | 44.3 | 64.1 | 52.4 | 43.0 | 63.3 | 51.2 | 37.9 | 53.1 | 44.2 |
| Pix2Plan w/o JM | 64.2 | 44.1 | 62.7 | 51.8 | 42.7 | 61.8 | 50.5 | 37.7 | 52.2 | 43.7 |
| Pix2Plan (full) | **65.0** | **45.4** | **66.6** | **54.0** | **43.8** | **65.6** | **52.5** | **38.7** | **54.9** | **45.4** |

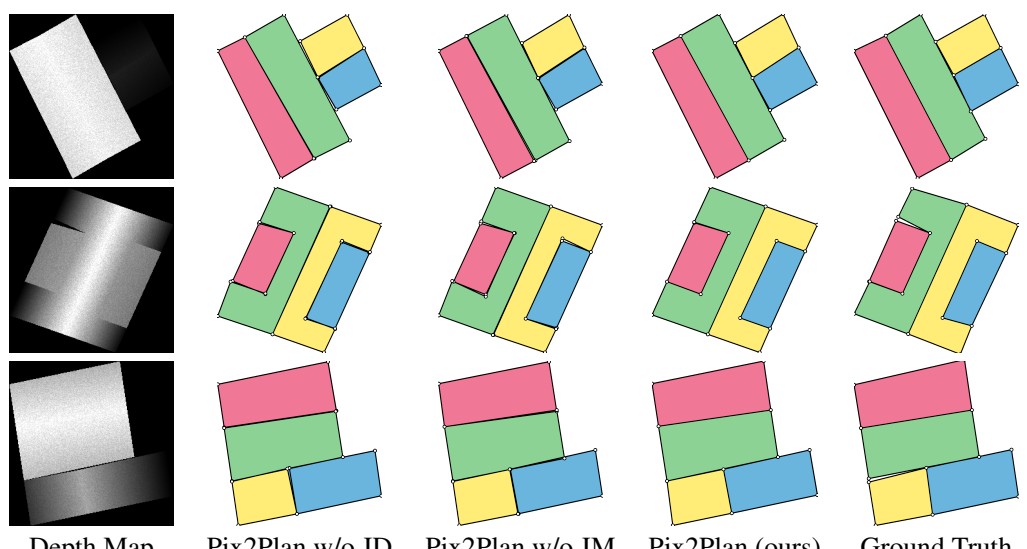

| Depth Map | Pix2Plan w/o JD | Pix2Plan w/o JM | Pix2Plan (ours) | Ground Truth |
|---|---|---|---|---|

Figure 5: **Qualitative results of the ablation experiments on Building3D (Wang et al., 2023) dataset.** 'Pix2Plan w/o JD' denotes the version of Pix2Plan without the Junction Decoder. 'Pix2Plan w/o JM' denotes the version of Pix2Plan without the Vertex-Junction Matching step. 'Pix2Plan (full)' is the full architecture with both the Junction Decoder and Vertex-Junction Matching step. Roof segment colors are assigned randomly and do not indicate any class labels.

are also common in preceding methods such as RoomFormer (Yue et al., 2023), FRI-Net (Xu et al., 2024) and PolyRoom (Liu et al., 2025). Despite these errors, it should also be noted that Pix2Plan achieves state-of-the-art performance for the task of building roof wireframe extraction and matches the state-of-the-art single-stage methods for the task of indoor floorplan extraction.

## D  ADDITIONAL QUALITATIVE EXAMPLES

In this section, we illustrate additional qualitative examples of building roof wireframes and indoor floorplans predicted by Pix2Plan in Figures 7 and 8 respectively.

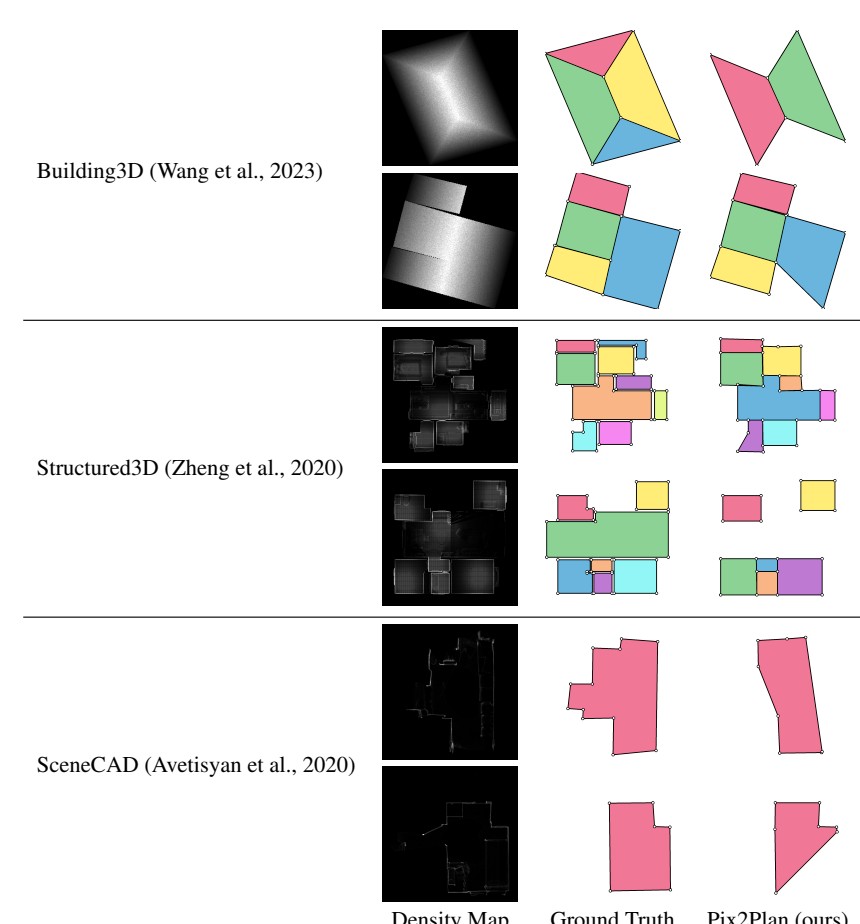

Figure 6: **Qualitative examples of failure cases of Pix2Plan on various datasets.** The most common modes of failure are when the network partially or completely misses roof/room polygons.

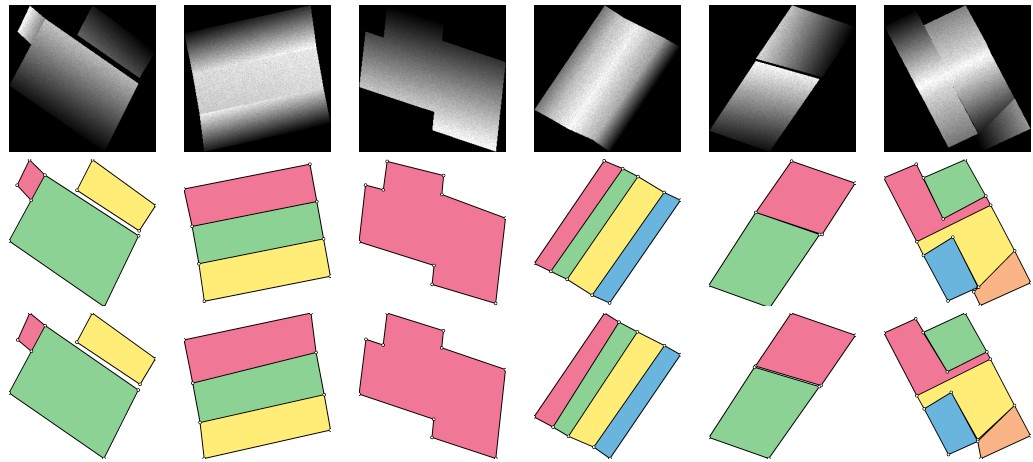

Figure 7: **Qualitative examples of building roof wireframes predicted by Pix2Plan on the Building3D (Wang et al., 2023) test set.** The first row depicts the input depth maps, second row depicts the corresponding ground truth building roof wireframe, and the third row showcases Pix2Plan's building roof predictions. Pix2Plan is able to consistently predict accurate building roof wireframes in a tight-layout.

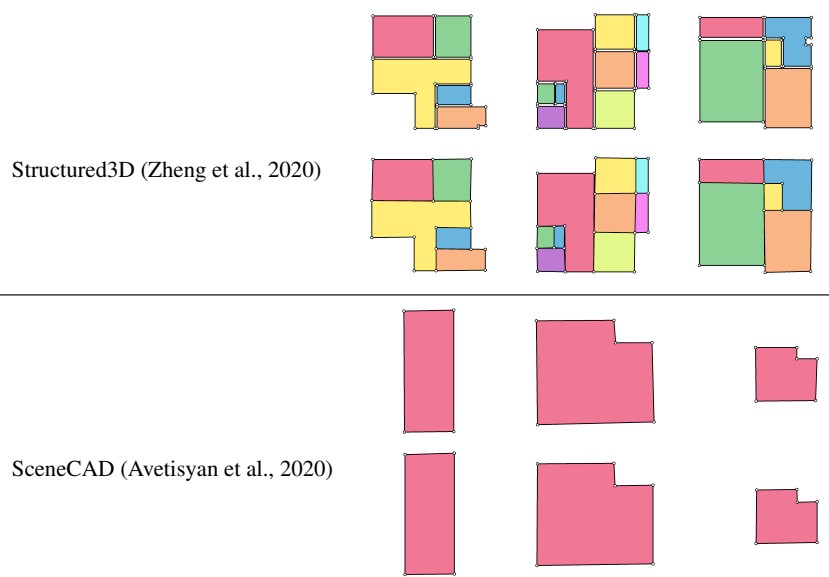

Structured3D (Zheng et al., 2020)

SceneCAD (Avetisyan et al., 2020)

Figure 8: **Qualitative examples of indoor floorplans predicted by Pix2Plan on the Structured3D (Zheng et al., 2020) test set and the SceneCAD (Avetisyan et al., 2020) validation set.** For each dataset, the first row depicts the ground truth indoor floorplan and the second row showcases Pix2Plan's floorplan predictions. Pix2Plan is able to consistently predict accurate indoor floorplans in a tight-layout.

