# OpenReview forum: "Pix2Plan: A Set Prediction Approach for End-to-End Wireframe Parsing using Two-Level Polygon Queries"
_ICLR.cc/2026/Conference — ICLR 2026 Conference Withdrawn Submission_

### Official Review · Reviewer_p5ry · 2025-10-18

**Soundness:** 2
**Presentation:** 3
**Contribution:** 2
**Rating:** 4
**Confidence:** 4

**Summary:**

The paper introduces Pix2Plan, a single-stage, end-to-end transformer model for wireframe parsing of built environments from remotely sensed data such as LiDAR or RGB-D scans. By jointly predicting polygons and junctions through two-level polygon queries and a junction decoder, and aligning them via a vertex-junction matching module, Pix2Plan effectively integrates the strengths of both bottom-up and top-down methods to produce tight-layout planar graphs for roofs and floorplans.

**Strengths:**

- The combination of polygon- and junction-level predictions through vertex-junction matching effectively unifies polygon detection and planar graph construction, reducing the need for post-processing.

- The paper provides a balanced and insightful review of bottom-up and top-down approaches, clearly motivating the necessity of a hybrid model.

**Weaknesses:**

- While the idea of combining polygon and junction prediction is interesting, the core concept of employing polygon representations is not novel and the proposed models is largely builds upon existing DETR-style frameworks (e.g., RoomFormer, PQ-Transformer [1]) with relatively straightforward extensions.



- On Structured3D and SceneCAD, Pix2Plan underperforms compared to previous methods (as shown in Table 2,3), and the paper lacks a convincing explanation for this performance gap.

- The paper mentions high efficiency (line 25) but provides no runtime or memory comparison with baselines.

[1] Pq-transformer: Jointly parsing 3d objects and layouts from point clouds

**Questions:**

Pix2Plan performs worse than some previous methods on indoor datasets such as Structured3D and SceneCAD. It would be helpful if the authors could analyze the underlying reasons for this performance gap. Additionally, an efficiency comparison (in terms of inference time, parameter count, or training cost) with baselines would strengthen the paper’s claim of “highly efficiency”.

---

### Official Review · Reviewer_3t2B · 2025-10-29

**Soundness:** 3
**Presentation:** 3
**Contribution:** 2
**Rating:** 2
**Confidence:** 4

**Summary:**

This paper introduces a single-stage transformer architecture for 2D wireframe and floorplan parsing, which employs two dedicated decoders to detect corner junctions and segment polygons, respectively.
Each predicted polygon vertex is then attached to its nearest corner junction, thereby forming a tight and compact wireframe or floorplan layout.
Evaluations on three standard public benchmarks somewhat demonstrate the effectiveness of the proposed method.

**Strengths:**

1. Driven by the goal of generating compact wireframe layouts, the authors combine corner junction and polygon segment detection in a single-stage transformer architecture.
2. Inter-primitive connectivity is achieved by assigning each polygon vertex to its nearest corner junction, yielding a more compact and structured layout representation.

**Weaknesses:**

1. The technical contribution appears limited. The DETR-based architecture of junction detection and polygon prediction using two-level queries is somewhat similar to the approach established in *HEAT* and *RoomFormer*. Moreover, as indicated by `Tab.4`, integrating the junction decoder alongside polygon decoder during training adversely leads to a performance degradation. Besides, the combination of corner junction and polygon segment detection through the Vertex-Junction Matching module is kind of trivial.
2. While the method achieves satisfactory performance on *Building3D* in `Tab.1`, its overall performance is limited. Specifically, evaluation results on floorplan reconstruction benchmarks (*Structured3D* in `Tab.2` and *SceneCAD* in `Tab.3`), do not demonstrate a clear superiority over existing methods.
3. Beyond the quantitative results, the paper would benefit from a more explicit elaboration of the other advantages offered by its integrated junction-polygon architecture, which may be evaluated through additional application-specific metrics. Please also refer to `Q2`.
4. A comparison with *HEAT* on *Building3D* in `Tab.1` is missing. It is essential since the proposed method builds upon the architectures of both *HEAT* and *RoomFormer*, yet a comparison with *HEAT* is omitted for this specific dataset.
5. Minor concern: the design of two dedicated decoders likely results in lower computational efficiency compared to more streamlined, single-decoder architectures.

**Questions:**

1. Several normalized depth maps in `Fig.2` appear weird,  exhibiting noticeable deformations.
2. The ground-truth annotations in the evaluated datasets do not consistently produce the desired "tightness". I am wondering if this hinders the superior performance of the proposed method in terms of quantitative evaluation. Maybe the authors could resort to a more targeted evaluation that better reflects the method's advantage.

---

### Official Review · Reviewer_5UK2 · 2025-10-31

**Soundness:** 3
**Presentation:** 3
**Contribution:** 3
**Rating:** 2
**Confidence:** 4

**Summary:**

The paper focuses on the task of extracting accurate wireframes of built environments and introduces a method called Pix2Plan. The method employs a DETR-style encoder-decoder transformer to predict a set of two-level polygon queries and a global set of junction vertices.

**Strengths:**

1. A method for extracting accurate tight-layout wireframes from remotely sensed data;

**Weaknesses:**

The paper focuses on the task of extracting accurate wireframes of built environments and introduces a method called Pix2Plan. The main drawback is that the method's novelty of design and method's performance are not significant enough.

1. The novelty compared to existing DETR-style methods is not significant enough.

2. The method's performance is not as good as the comparison methods on Structured3D or SceneCAD, according to Tables 2-3. It cannot convince us of Pix2Plan's superiority.

3. The method's ablation study in the Appendix is about the Junction Decoder and Vertex-Junction Matching step. It's hard to say the contributions of the two designs. Once again, both the method novelty and the performance of the method might not be significant.

4. Comparison of the training and testing computation statistics can be provided for further discussion.

**Questions:**

As above.

---

### Official Review · Reviewer_tigV · 2025-10-31

**Soundness:** 3
**Presentation:** 3
**Contribution:** 2
**Rating:** 2
**Confidence:** 4

**Summary:**

This paper presents a single-stage transformer-based method for extracting building roof and indoor floorplan wireframes from remotely sensed data (density map). The approach combines polygon-level detection with junction prediction to produce tight-layout planar graphs. The method reads sound, but it seems the performance does not exceed the prior arts, especially these two-stage methods.

**Strengths:**

Although the performance is not as good as two-stage methods, the vertex-junction matching produces planar graphs without complex post-processing, which is a practical advantage for downstream CAD applications.

Testing on three datasets (Building3D, Structured3D, SceneCAD) with multiple metrics provides thorough validation.

**Weaknesses:**

The major weaknesses come from the limited novelty and incompetent performance.

1. The main contribution is essentially adding a junction decoder and distance-based matching, and the architecture heavily borrows from existing work (DETR, Deformable DETR, RoomFormer's two-level queries). The matching step is quite simple (nearest neighbour).

2. It underperforms two-stage methods significantly on Structured3D and SceneCAD.

**Questions:**

Why does the method excel on Building3D but lag on floorplan datasets? Is this architectural or task-specific?

Can the junction decoder be used to improve two-stage methods like PolyRoom?

It seems the ablation study ablates the junction decoder components. Every contributive component claimed by the authors should be ablated.

In section3.2, how are duplicate matches handled when multiple polygon vertices match the same junction? And what happens when junctions have no nearby polygon vertices?

---

### Note · Authors · 2025-11-14

**Comment:**

We thank the reviewers for their valuable feedback. Considering the comments received, we would like to withdraw our submission and work on the suggested improvements.

**Withdrawal Confirmation:**

I have read and agree with the venue's withdrawal policy on behalf of myself and my co-authors.